# Pharmacologic ROMK Inhibition Protects Against Myocardial Ischemia Reperfusion Injury

**DOI:** 10.3390/ijms26083795

**Published:** 2025-04-17

**Authors:** Allison C. Wexler, Holly Dooge, Lara Serban, Aditya Tewari, Babak M. Tehrani, Francisco J. Alvarado, Mohun Ramratnam

**Affiliations:** 1Division of Cardiovascular Medicine, Department of Medicine, University of Wisconsin School of Medicine and Public Health, Madison, WI 53705, USA; acwexler@wisc.edu (A.C.W.); hdooge@wisc.edu (H.D.); lserban@student.ethz.ch (L.S.); atewari3@wisc.edu (A.T.); btehrani@wisc.edu (B.M.T.); falvarad@medicine.wisc.edu (F.J.A.); 2Cardiology Section, Medical Service, William S. Middleton Memorial Veterans Hospital, Madison, WI 53705, USA; 3Cardiovascular Research Center, University of Wisconsin School of Medicine and Public Health, Madison, WI 53705, USA

**Keywords:** K_ATP_ channel, renal outer medullary potassium channel, myocardial ischemic reperfusion injury, mitoK_ATP_

## Abstract

Mitochondrial ATP-sensitive K+ channels are closely linked to cardioprotection and are potential therapeutic targets during ischemia reperfusion (IR) injury. The renal outer medullary K+ channel isoform 2 (ROMK2) is an ATP-sensitive K+ channel found in the mitochondria of cardiomyocytes. While the germline knockout of ROMK does not mediate myocardial IR injury, the effect of ROMK loss of function on IR injury in the adult myocardium is unknown. By using a selective small molecule inhibitor of ROMK, we paradoxically found that mouse hearts were protected from IR injury after ROMK inhibition compared to vehicle-treated animals. In addition, we found that ROMK inhibition leads to exaggerated mitochondrial uncoupling and increased ROS production. Phosphatidylinositol 4,5-bisphosphate (PIP_2_), an activator of ROMK, increased the effect of ATP to hyperpolarize cardiac mitochondrial membrane potential. ROMK inhibition also increased mitochondrial swelling in the absence of ATP. In conclusion, pharmacologic ROMK inhibition protects the murine heart from IR injury and may promote a phenotype of enhanced mitochondrial matrix K+. ROMK may be more important during conditions that promote mitochondrial matrix K+ efflux than influx. Further research to understand its role in mitochondrial K+ handling and as a therapeutic target in IR injury is needed.

## 1. Introduction

Therapies targeting mitochondrial ATP-sensitive K+ channels (mitoK_ATP_) to treat myocardial ischemia reperfusion (IR) injury, while promising, continue to be a challenge for clinical translation. ATP-regulated K+ channels (K_ATP_) couple the cellular or organelle energetic status to membrane excitability [1,2]. K+ is the most abundant cation in the cytosol and is also thought to be present in high concentrations (~150 mM) within the mitochondrial matrix [3,4]. MitoK_ATP_ channels, present in the inner membrane of mitochondria, open in response to a reduced ATP/ADP ratio and subsequently increase K+ flux into the mitochondrial matrix. This leads to enhanced mitochondrial swelling and improvement in energy transfer [5,6,7]. Cardiomyocytes benefit significantly from this physiological process as enhanced mitoK_ATP_ activity is associated with ischemic preconditioning and cardioprotection after IR injury [8,9,10,11].

Since the initial discovery of a mitoK_ATP_ channel in 1990 [12], major strides in identifying the structural components of the channel continue. Investigators recently identified an ATP-sensitive mitochondrial K+ channel composed of CCDC51 and a regulatory accessory protein, ABCB8 [13]. This channel resides in the inner mitochondrial membrane and is associated with diazoxide-induced cardioprotection. Prior to the discovery of CCDC51, the renal outer medullary K+ channel isoform 2 (ROMK2), encoded by *KCNJ1*, was the leading candidate for the K+ pore of mitoK_ATP_ [14]. ROMK is an inward rectifying K+ channel extensively studied in the kidney [15]. These splice variants exhibit identical biophysical properties but harbor different NH_2_-terminal amino acid sequences. Studies performed in lipid bilayers, mitoplasts, cardiomyocytes, and isolated rat heart tissue found ROMK2 to be a mitochondrial ATP-sensitive K+ channel [14,16,17]. In addition, we found that ROMK is able to regulate mitochondrial membrane potential and uncoupling in isolated adult murine cardiomyocytes [18]. While evidence suggests a role for ROMK2 in mitochondrial K+ regulation, recent investigations in mice with germline cardiac ROMK2 loss of function moved ROMK2 away from being a mitochondrial K+ influx channel [19]. However, it is unclear if these results may have resulted from compensatory changes during development and what effects acute perturbation in ROMK function has on the adult myocardium.

Another element of the ROMK story is the potential value of ROMK inhibitors as therapeutic agents in hypertension and cardiovascular disease. ROMK is highly expressed in the kidney and plays a significant role in ion homeostasis [15]. Its relevance to human physiology and disease was first described in the kidney, where the loss of function mutations lead to renal tubular salt-wasting and Barter’s syndrome [20]. Subsequently, researchers found that genetic variation in *KCNJ1* is associated with lower blood pressure and that ROMK may be a potential therapeutic target for hypertension [21,22]. Ongoing efforts by pharmaceutical companies and academia to produce inhibitors of ROMK for the treatment of hypertension continue. Therefore, it is important to understand the effects of ROMK inhibitors outside of the kidney and especially the heart for future translation efforts.

We have recently acquired a novel ROMK2 inhibitor developed by MERCK & Co Inc.—Compound A. Compound A is a selective ROMK inhibitor with specificity over other K+ channels at concentrations as high as 100 µM [23] in vitro. Studies have shown that Compound A affects kidney physiology and ion homeostasis with a beneficial effect on hypertension [24].

Therefore, in this study, we use a pharmacologic approach to assess the significance of ROMK inhibition on myocardial IR injury and mitoK_ATP_ activity.

## 2. Results

### 2.1. Pharmacologic ROMK Block Improves Infarct Size in Mice After In Vivo IR Injury

To determine if ROMK inhibition may influence the response to myocardial IR injury, C57BL/6J mice were treated with either vehicle (DMSO solution) or the ROMK inhibitor Compound A (3 mg/kg/day) starting 24 h prior to the induction of IR injury (Figure 1A). Mice were subjected to 45 min of ischemia followed by 72 h of reperfusion. We found that treatment with the ROMK inhibitor significantly decreased infarct size, our primary outcome, as measured by the percentage of infarct area to area at risk (Figure 1B,C). In addition, there was a survival benefit in mice treated with the ROMK inhibitor compared to vehicles (Figure 1D). ROMK inhibition led to a reduction in LV end diastolic dimension compared to both vehicle-treated and sham controls. In addition, while there was no change in cardiac function between ROMK inhibitor treatment and sham controls, there was a non-significant trend toward improved function in ROMK inhibitor-treated compared to vehicle-treated mice after IR injury (%FS 34% ± 8% vs. 30% ± 7%, *p* = 0.1; EF 64% ± 11% vs. 58% ± 10%, *p* = 0.01, Figure 1E). The increased survival in ROMK inhibitor-treated mice despite no significant reduction in LV ventricular performance between groups may be attributed to increased arrhythmias in the vehicle group. This was further examined in isolated heart preparation below. The reduction in LV volumes from ROMK inhibition likely stems from the fact that ROMK inhibitors are diuretics, and Compound A is known to produce a diuresis in rodents at the dose used [23]. However, diuretics are not associated with an improvement in infarct size or survival after myocardial infarction. Therefore, there may be non-renal and cardiac-specific effects of ROMK inhibition at play that are influencing the response to myocardial IR injury.

### 2.2. Pharmacologic ROMK2 Block Protects the Heart During Ex Vivo Myocardial IR Injury

In order to investigate whether cardiac-specific protection during myocardial IR injury may be afforded by ROMK2 inhibition, the isoform found in cardiac tissue, we employed an isolated heart perfusion IR injury model. In addition, given the large evidence base that mitochondrial ATP-sensitive K+ channels are linked to ischemic preconditioning (IPC) [25,26], we employed this ex vivo model to study the effect of ROMK2 inhibition on IPC-induced protection. Baseline ex vivo hemodynamic parameters between vehicle and ROMK2 inhibition perfused mouse hearts are presented in Table 1.

While there were no statistically significant differences between groups, there was a trend toward a lower spontaneous heart rate and systolic (+dp/dt) and diastolic (−dp/dt) function. Therefore, epicardial pacing was used to match heart rates between groups, and % pressure recovery was used to assess hemodynamic response between groups. We found that isolated mouse hearts perfused with the ROMK inhibitor at 30 µM demonstrated improved recovery of left ventricular pressure, +dp/dt, and −dp/dt compared to vehicle treatment (Figure 2). We also found, similar to our in vivo experiments, that ROMK inhibitor-perfused hearts have lower infarct size compared to vehicle-treated mice (Figure 3). Ischemic preconditioning in vehicle-perfused mice had improved hemodynamic recovery and infarct size compared to vehicle-perfused mice without ischemic preconditioning (Figure 2 and Figure 3). ROMK inhibitor perfusion, however, blocked the effect of ischemic preconditioning, with no significant improvement in the recovery of LVDP, +dp/dt, −dp/dt, and infarct size compared to vehicle-perfused hearts without IPC (Figure 2 and Figure 3). We also found a reduction in ventricular ectopic beats between the vehicle-perfused group compared to the ischemic preconditioning and ROMK inhibitor-perfused group (Appendix A).

### 2.3. ROMK2 Inhibition Enhances Flavoprotein Oxidation in Cardiomyocytes

Our results were surprising and suggest a role for cardiac ROMK2 in mediating IR injury. Since ROMK2 has been found in mitochondria [14,15] and functions as an ATP-sensitive inward rectifying K+ channel in the kidney [15], we explored assays to assess mitoK_ATP_ activity in isolated adult cardiomyocytes. We have previously reported that ROMK2 inhibition paradoxically leads to enhanced uncoupling in adult murine cardiomyocytes, especially in the presence of diazoxide, a mitoK_ATP_ activator [18]. Increasing mitochondrial matrix K+ can lead to uncoupling of mitochondria. Therefore, we hypothesize that ROMK2 has a greater impact on mitochondrial K+ efflux than influx when the environmental conditions favor K+ exit from the mitochondrial matrix. This is based on the channel’s inherent inward rectifying properties (inward current directed into the cytoplasm in this case) and that blockade of ROMK2 may enhance mitochondrial matrix K+ and thus lead to uncoupling. The direct measurement of mitochondrial K+ in intact cells is challenging with current methods. The K+-sensitive probes, BTC or PBFI, require cell permeabilization or non-physiologic buffers [27].

Therefore, we turned to FAD autofluorescence as an intact cellular measure for mitoK_ATP_ activity [10,25]. The uncoupling induced by mitoK_ATP_ activation can be sensed by a rise in FAD autofluorescence. We found that isolated adult murine cardiomyocytes treated with a ROMK inhibitor had a higher resting FAD autofluorescence compared to vehicle-treated cells (Figure 4A,B). Some prior investigators have found that diazoxide increases FAD autofluorescence in cardiomyocytes cultured overnight and incubated in a buffer free of metabolic substrates [10], while others have not found a significant effect with diazoxide [28]. When we perfused cells with diazoxide, we found no significant change in FAD autofluorescence. Since our aim was to assess conditions in a physiology buffer, we assessed the effect of diazoxide on FAD autofluorescence with ROMK inhibitor-treated cells. We observed a small but significant rise in FAD autofluorescence when cardiomyocytes were treated with the ROMK inhibitor after perfusion with diazoxide (Figure 4E). This again suggested that cardiac ROMK2 inhibition may increase mitochondrial matrix K+ and produce a cellular phenotype similar to mitoK_ATP_ activation than inhibition.

### 2.4. ROMK2 Inhibition Increases Mitochondrial ROS Production at ETC Complex III and Reduces ROS Production at Complex I

While our current data with FAD autofluorescence and our past work studying mitochondrial membrane potential [18] suggest ROMK2 inhibition enhances mitochondrial uncoupling in cardiomyocytes, another potential mechanism of how ROMK2 inhibition leads to cardioprotection may be via ROS generation. The relationship between ROS production and protection from IR injury is complex. While many reports suggest that IPC and mitoK_ATP_ activation results in ROS production [11,29], a significant body of literature exists supporting the role of increased ROS production in reperfusion leading to irreversible myocardial damage [30]. To reconcile these findings, the field postulates that activation of mitoK_ATP_ leads to the generation of signaling ROS, which promotes a cardioprotective phenotype, ultimately suppressing reperfusion ROS [31]. In order to investigate the effect of ROMK2 blockade on ROS production, we loaded isolated cardiomyocytes with the mitochondrial superoxide-sensitive dye mitoSOX red and perfused cells with hydrogen peroxide. Treatment with H_2_O_2_ increases mitochondrial superoxide production due to an eventual reduction in the defenses used to combat ROS production. Isolated adult murine cardiomyocytes treated with the ROMK inhibitor have an increase in superoxide production compared to vehicle-treated cells (Figure 5A,B). This result may be consistent with increased ROS signaling production that is also observed during IPC and activation of mitoK_ATP_ [11]. We further tested ROS production at different ETC sites. We found that cells treated with antimycin, a complex III inhibitor, had a small but statistically significant rise in ROS production after ROMK inhibitor treatment, while cells treated with rotenone, a complex I inhibitor, had a statistically significant reduction in ROS production (Figure 5C,D). ROS production at complex III may release superoxide anion into the intermembrane space, thus acting as a signaling molecule to the cytoplasm [32]. This is in direct contrast to ROS production at complex I, which increases mitochondrial matrix ROS and can lead to detrimental effects such as mitochondrial transition pore opening [33]. Therefore, ROMK2 inhibition may increase the production of signaling ROS similar to IPC while also decreasing the production of harmful ROS.

### 2.5. PIP_2_ and pH Affect the ATP Sensitivity of Cardiac Mitochondrial Membrane Potential

Next, we attempted to measure changes in mitoK_ATP_ activity by the assessment of mitochondrial membrane potential (ψm) in isolated mitochondria. Prior reports suggest that mitoK_ATP_ activation leads to small changes in ψm, making observations challenging in cell-based techniques. In our prior report, we did not see a significant effect in ψm from isolated cardiomyocytes with the mitoK_ATP_ activator Diazoxide or ROMK2 inhibition [18]. Therefore, we turned our attention to the measurement of ψm in isolated mitochondria to assess mitoK_ATP_ activity in the presence of complex I substrates and blockade of ATP synthase by oligomycin. First, we found that ATP hyperpolarizes cardiac ψm in a dose-dependent manner with an IC_50_ of 201 µM (Figure 6A,B). In kidney cells, ROMK is in a more active state at higher pH [34] and is activated by the cofactor Phosphatidylinositol 4,5-bisphosphate (PIP_2_) [35,36]. In addition, mitoK_ATP_ is regulated by PIP_2_ [37].

Therefore, we then assessed the ATP sensitivity of ψm with the addition of PIP_2_ and varying pH. We hypothesize that mitochondrial ROMK2 functions as an inward rectifying channel with the intermembrane space/cytoplasmic facing side of the site of regulation. We found that the addition of PIP_2_ increased the sensitivity of ψm to lower concentrations of ATP and produced a curve with a better dose–response fit and a reduced IC_50_ of 36 µM (Figure 6A,B). Interestingly, the peak inhibitory effect of ATP on ψm hyperpolarization and mitoK_ATP_ inhibition was diminished in PIP_2_-treated mitochondria and occurred at a lower ATP level of 300 µM, suggesting PIP_2_ was acting like a partial agonist with an increase in ATP potency but a reduction in maximum efficacy (Figure 6C). In the kidney, ROMK activity is sensitive to pH, with higher pH levels leading to an increased probability of the open state [34]. We found that a higher pH in our mitochondrial preparations leads to increased ATP sensitivity compared to a more acidic pH (Figure 7). The ATP sensitivity of ψm with PIP_2_ and pH regulation continued to support the physiological role of ROMK2 in mitochondria.

### 2.6. ATP Sensitivity of Cardiac Mitochondria Is Altered by ROMK2 Inhibition Only in the Setting of PIP_2_

Next, we tested whether the ATP sensitivity of isolated mitochondria is altered with ROMK2 inhibition. We tested the effect of different concentrations of the ROMK2 inhibition on the ATP sensitivity of ψm but were unable to detect a significant difference in comparison to vehicle-treated preparation (Appendix A). However, when we added PIP_2_ to the preparations, we found that ROMK2 inhibition paradoxically increased the ATP sensitivity of ψm (Figure 8A,B). This provided more support that ROMK2 is an ATP-sensitive Kir channel in mitochondria. However, the observed results suggest that ROMK2 may be important in mitochondrial K+ efflux.

### 2.7. Pharmacologic ROMK2 Blockade Alters Swelling in Isolated Cardiac Mitochondria

To further study the effect of ROMK2 inhibition on mitoK_ATP_ activity, we used the mitochondrial matrix swelling assay in isolated cardiac mitochondria from mice [38]. When mitochondria increase K+ uptake, the matrix swells by the concomitant osmotic entry of H_2_O. This is evident with the addition of valinomycin, a K+ ionophore, which significantly increases mitochondrial swelling in our preparations, while treatment with FCCP, an H+ ionophore, does not impact mitochondrial swelling (Appendix A). In mitochondrial preparations treated with vehicle alone, we saw a non-significant increase in mitochondrial swelling of 14% between preparations in 0 µM compared to 1000 µM ATP (*p* = 0.08, Figure 8C). The lack of observed statistical significance may be due to channel run down despite us using mitochondrial within 2 h of isolation for each experiment, a known phenomenon when working with Kir channels [37]. However, we found significant differences in swelling with preparations treated with the ROMK inhibitor between 0 µM and 1000 µM ATP. Doses of 1, 3, and 10 µM ROMK inhibitor produced a significant swelling increase in mitochondria of 24%, 10%, and 12%, respectively (Figure 8C). Our results using the ROMK inhibitor on isolated mitochondria suggested that mitochondrial ROMK2 mediates the K+ cycle and may behave in concert with CCDC51 to regulate mitochondrial K+ in times of energy depletion.

## 3. Discussion

We found that the pharmacologic inhibition of ROMK promotes cardioprotection after myocardial IR injury and that cardiac ROMK2 inhibition enhances a phenotype of increased mitoK_ATP_. These conclusions are based on the following evidence. First, we demonstrated that pharmacologic ROMK inhibition promoted cardioprotection after myocardial IR injury in mice in vivo and ex vivo. Next, we provided evidence that ROMK2 inhibition in isolated adult murine cardiomyocytes enhances mitochondrial uncoupling and ROS production. Finally, we provide data that ROMK inhibition leads to an increase in the ATP sensitivity of ψm in isolated cardiac mitochondria. Collectively, this experimental evidence supports the overall concept that cardiac mitochondrial ROMK2 may act preferentially as a mitochondrial K+ efflux channel, and inhibition may increase matrix K+, leading to cardioprotection (Figure 9).

ROMK2 was originally found in heart tissue in 2012 by Dr. O’Rourke’s laboratory as they searched for the K+ pore for mitoK_ATP_ [14]. It was subsequently found in mitochondria in dermal fibroblasts and human embryonic stem cell-derived cardiomyocytes [17,39]. These initial publications increased excitement that ROMK2 was the K+ pore for mitoK_ATP,_ with subsequent experiments showing ROMK2 functioning as a mitochondrial ATP-sensitive K+ channel [16]. However, recent evidence has moved ROMK2 away from the K+ pore of mitoK_ATP_. First, Paggio et al. recently provided strong evidence for a channel composed of CCDC51 and ABCB8 as the mitoK_ATP_ channel [13]. While this discovery did not rule out the possibility of an alternative mitoK_ATP_-ROMK-based channel, subsequent research found that cardiac-specific ROMK2 knockout mice did not fare differently after myocardial IR injury than WT mice [19]. While these data appear discordant with our findings, there are notable points that may tie many of the discrepancies together. It may be that compensatory changes have occurred in the ROMK2 conditional KO model previously described. This was the case when researchers sought to understand the role of the mitochondrial calcium uniporter (MCU) in IR injury by using genetically altered rodent models. While global MCU knockout mice were found to have no significant difference in response to myocardial IR injury compared to WT controls, the inducible conditional mice with a deletion in adulthood were found to fare better after IR injury [40,41]. Thus, mitochondria may have compensatory mechanisms for K+ homeostasis, similar to the regulation of Ca^2+^. Therefore, our pharmacologic model may circumvent any compensatory changes in mitochondrial K+ regulation from germline ROMK deletion.

In the field of mitochondrial K+ channel research, significant effort over the last three decades to identify the K+ uniporter has culminated in the discovery of CCDC51 and its accessory protein ABCB8. This complex behaves as the major K+ influx channel. However, there is still a lack of knowledge regarding significant efflux pathways for K+ in mitochondria. The current model originally proposed by Mitchell was the existence of a K+/H+ antiporter to prevent excessive matrix K+ and there is experimental evidence for its existence [42]. However, the molecular structure of such a transporter has yet to be identified. ROMK was discovered in 1990 as a founding member of the inward rectifying K+ channel family. These channels are more likely to conduct K+ into the cell, given favorable environmental conditions. The pore of the channel is blocked by Mg^2+^ or polyamines that are removed during conditions that promote inward rectification [15]. Alternative splicing of the ROMK gene, Kir1.1, produces three different isoforms: ROMK1, ROMK2, and ROMK3. Their biophysical properties appear to be similar, and they differ in NH_2_-terminal amino acid sequence. ROMK2 was found to be the predominant isoform expressed in heart tissue [14]. Our data suggest that the inhibition of ROMK2 leads to heightened mitoK_ATP_ channel activity. In addition, alterations in pH and the addition of PIP_2_ or a ROMK inhibitor to the buffer in preparations of isolated cardiac mitochondria affect channel function, supporting regulation on the cytoplasmic side. Prior studies in mitoK_ATP_ research have suggested regulation of the channel from the cytoplasmic side [43]. Thus, we propose that the inhibition of ROMK2 may enhance K+ flux into the matrix, which is performed by the CCDC51/ABCB8 complex.

Our results suggest a role for ROMK2 in mitochondrial K+ handling. The inhibition of cardiac ROMK2 leads to downstream effects that promote cardioprotection. We have previously shown that ROMK inhibition enhances cardiac mitochondrial uncoupling, a known phenotype that promotes cardioprotection [44]. In addition, ROS generation is complex in mitochondria during IR injury. MitoK_ATP_ activation increases mitochondrial ROS production [45]. The production of this signaling ROS may then activate downstream protective pathways [46]. We found a rise in ROS at complex III with a reduction in ROS production at complex I. The superoxide anion produced at complex III is liberated into the intermembrane space and can function as signaling ROS [32]. While ROS produced by complex I stay in the mitochondrial matrix and can lead to cell death via MPTP opening [33]. There may also be unidentified sites of signaling ROS production from increases in matrix K+ due to the inhibition of ROMK2. Another interesting finding we observed is that ischemic preconditioning is unable to further protect hearts after ROMK inhibition. Hearts that were treated both with IPC and ROMK inhibition had no change in hemodynamic recovery and infarct size compared to mice that had no treatments. This may be due to a critical point of K+ overload in the mitochondrial matrix and excessive swelling as both IPC and ROMK inhibition, we hypothesize, are leading to mitochondrial K+ accumulation. Prior studies with valinomycin, a K+ ionophore, showed significant swelling of mitochondria from unabated K+ entry, leading to membrane rupture and cytochrome c release [5]. The concomitant treatment of hearts with IPC and ROMK inhibition may reach a threshold of too much mitochondrial matrix K+, switching the balance between protection to death.

In order to tie the ROMK2 chapter with the mitochondrial K+ cycle story, future studies in inducible cardiac-specific ROMK2 knockout mice would be needed. This model more closely resembles our pharmacologic model and has a better chance of circumventing any compensatory changes that may occur during development. In addition, while our experiments found a role for ROMK2 inhibition in IR injury, future studies to test the therapeutic potential of ROMK2 are needed. Experiments devised to give ROMK2 inhibitors after the initiation of IR injury in vivo would be important for further clinical translation. In addition, the expression changes and subsequent activity of cardiac ROMK2 in different forms of human heart disease are unknown. Further research understanding the effects that chronic heart failure has on ROMK2 may also enhance clinical translation. Finally, the precise mechanisms for how ROMK2 may behave preferentially as a K+ efflux channel in mitochondria are needed. Since mitochondria have a high positive potential in resting conditions, the movement of K+ greatly favors the inward direction. While Mg^2+^ and polyamines, known to block Kir channels, as well as ATP regulation, may inhibit the inward flux of K+ through ROMK2, how the channel ultimately leads to K+ efflux is uncertain. In ischemic conditions, when K+ rises in mitochondria due to CCDC51 activation, mitochondrial membrane potential falls, and there is a loss in cytoplasmic K+ [47], an environment may occur that favors mitochondrial K+ efflux via ROMK2. In addition, similar to mitochondrial Ca^2+^ entry, there may be microdomains [48] of higher K+ concentration, resulting in a driving force for K+ exit. This could be in areas where CCDC51 and ROMK2 are in close proximity. ROMK2 may also couple with a H+ transporter—but we find this unlikely. If ROMK2 were part of the K+/H+ antiporter complex, we would have thought this would lead to a significant baseline cardiac phenotype due to enhanced mitochondrial matrix K+ and swelling both in patients with ROMK2 loss of function mutations and cardiac-specific ROMK2 KO mice. We surmise that ROMK2 is not constitutively active and participates in K+ regulations during conditions that enhance K+ entry into mitochondria. Finally, our prior results have also found that the regulation of ROMK2 may be linked to a small splice variant of the sulfonylurea receptor (we termed SUR2A-55 [18]). Thus, while cellular stress such as ischemia may induce mitochondrial K+ influx via a CCDC51/ABCB8 channel, we propose this is counterbalanced by a ROMK2/SUR2A-55 channel promoting K+ efflux. However, further work is needed to support this claim in more specific models.

While we believe our experimental results support a role for ROMK2 in mitochondrial K+ handling and as a potential target for IR injury, there are limitations to our study. First, we used a pharmacological model, and off-target effects may have led to our results. Several ROMK2 inhibitors now exist. We chose to use Compound A, which has been shown to be specific to ROMK over other K+ channels [49] and was also used in rodents showing ROMK inhibition [23,24]. Regardless, future work in inducible ROMK2 KO mice would support our results and is planned. Second, the specific measurement of mitochondrial K+ is challenging. We used assays previously employed to assess mitoK_ATP_ activity, namely FAD autofluorescence, ROS generation, mitochondrial swelling, and mitochondrial membrane potential. While we found changes in these parameters with the use of our pharmacologic ROMK2 inhibitor, a direct measurement of mitochondrial K+ is lacking. One potential avenue for future research is the use of novel genetically encoded potassium indicators targeted to mitochondria [50].

## 4. Materials and Methods

### 4.1. Animal Usage

All animal procedures and experiments were performed with approval from the Institutional Animal Care and Use Committee at the University of Wisconsin. Male WT C57BL/6J mice ages 10–14 weeks were obtained from the University of Wisconsin Biomedical Research Model Services core for experiments. We are unaware of significant changes in ROMK2 activity or expression by sex and therefore used WT male mice for experiments. For each figure, the sample size is provided.

### 4.2. In Vivo Murine Myocardial IR Injury

Animals were randomly divided into 3 groups: Sham + vehicle treatment (DMSO), IR + vehicle treatment, and IR + ROMK2 inhibitor treatment. The ROMK2 inhibitor (Compound A, 3 mg/kg/day [24], MERCK & Co Inc., Rahway, NJ, USA) or vehicle was started 24 h prior to IR injury or sham and continued for 72 h after IR injury (Figure 1A). Prior studies using Compound A have shown excellent specificity and selectivity for ROMK [23,49]. To simulate myocardial IR injury, mice were first anesthetized with 3.25 mg/kg of extended-release buprenorphine SQ and subsequently given isoflurane via forced ventilation (induction 3–5%, maintained: 1–5%). Mice were placed on a heated operating table to maintain a body temperature of 37 °C. Animals were then orally intubated and ventilated. A left lateral incision was made in the fourth intercostal space to allow access to the heart. A 7-0 prolene suture was placed through the myocardium in the anterolateral wall around the left anterior descending artery. The suture was secured in place, and ischemia was confirmed by observing blanching of the distal circulation (ventricular apex). After 45 min, the suture was removed. The ribs and muscle layers were closed by absorbable sutures while evacuating the chest. The skin was closed by additional suturing. The mouse was then recovered from the anesthesia and extubated. Mice were monitored for 72 h post-surgery, at which time echocardiography was performed and hearts were harvested. Infarct sizes were determined by calculating the percentage of myocardial infarction compared to the area at risk (AAR) using a double staining technique with Evan blue and triphenyltetrazolium chloride by a blinded laboratory member. The area at risk and the infarct size were determined via planimetry using ImageJ software 1.54f, and the degree of myocardial damage was calculated as a percentage of infarcted myocardium from the AAR. All animals that were successfully extubated after the IR protocol were included in the analysis.

### 4.3. Echocardiography

Transthoracic echocardiography was performed by a blinded sonographer on male WT mice after sedation with 3% and maintenance with 1.5% isoflurane using a VisualSonics Vevo 3100 machine (Fujifilm, Lexington, MA, USA) with a 30 MHz transducer [51]. After ischemia (45 min) and reperfusion (72 h) injury. Left ventricular end diastolic (LVID;d) and left ventricular end systolic (LVID;s) were obtained from M-mode tracings. LV fractional shortening was calculated with the following equation: (LVEDD − LVESD)/LVEDD × 100%. LV ejection fraction was obtained from B-mode echocardiography.

### 4.4. Isolated Murine Heart IR Injury Experiments

Isolated mouse hearts were perfused according to the Langendorff technique at constant pressure, as previously described [51]. Mice, ages 10–14 weeks, were euthanized by cervical dislocation, and then hearts were quickly excised from the chest. The aorta was cannulated with a 22-gauge metal cannula, then secured with silk suture and hung on an isolated perfused heart system for small rodents (Harvard Apparatus, Holliston, MA, USA). Hearts were perfused at 37 °C at a constant pressure of 80 mmHg with a modified Krebs–Henseleit Buffer (118 mM NaCl, 22 mM NaHCO_3_, 4.7 mM KCl, 1.2 mM MgSO_4_, 1.1 mM KH_2_PO_4_, 2.5 mM CaCl_2_, 5 mM Glucose, 2 mM Sodium Pyruvate). The left auricle was then excised, and a fluid-filled balloon catheter constructed from a commercially available kit (Harvard Apparatus, Holliston, MA, USA) was placed in the left ventricle. The balloon catheter was attached to an APT300 pressure transducer (Harvard Apparatus, Holliston, MA, USA), and baseline left ventricular pressure was set between 5 and 10 mm Hg. Left ventricular pressure was recorded throughout the experimental protocol and analyzed using LabChart Pro (ADInstruments, Colorado Springs, CO, USA). Hemodynamic variables are presented as % recovery to baseline values. Global ischemia was induced by turning off flow to the isolated mouse heart. Hearts were subjected to 20 min of baseline perfusion and then either continued perfusion for 30 min or a stimulus of ischemic precondition with 3 cycles of ischemia followed by reperfusion in 5 min intervals (Figure 2A). Then, hearts were subjected to no flow global ischemia for 45 min, followed by 60 min of reperfusion. Hearts were paced at 360 bpm via epicardial pacing leads with a Grass SD9 stimulator during the last 55 min of reperfusion. In addition, mice were perfused with either vehicle (DMSO) or the ROMK2 inhibitor (Compound A 30 µM) throughout the experimental protocol, creating 4 experimental groups. IR experiments were excluded from analysis if (1.) the aortic block pressure or left ventricular developed pressure fell below 60 cm H_2_O during the initial 20 min perfusion, (2.) there was observed abnormally slow, fast, or arrhythmic beating, or (3.) there was an observed abnormal flow rate out of the heart during initial perfusion. After the IR injury protocol, hearts were removed from the Langendorff apparatus and arrested in diastole with 5 min of perfusion with 30 mM KCl in PBS. Hearts were then stained at 37 °C with 1% TTC in phosphate buffer (23.1 mM NaH_2_PO_4_, 76.9 mM Na_2_HPO_4_, 29.9 mM 2,3,5-Triphenyltetrazolium chloride TTC), frozen, and cut into 1 mm transverse slices. Sections were then imaged under light microscopy and photographed. Infarct size was determined through color threshold calculations in ImageJ software 1.54f (white pixels/pixels at risk) by blinded lab personnel. Ventricular ectopy was assessed by counting the aberrant ventricular beats during the first 5 min of reperfusion as previously described [52].

### 4.5. Isolation of Cardiomyocytes

Mice were dosed with an intraperitoneal injection of 200 units of heparin and returned to the carrier for 15 min before euthanasia by cervical dislocation. The heart was excised and dissected in a dish containing heparinized perfusion buffer (10 mM HEPES, 0.6 mM Na_2_HPO_4_, 113 mM NaCl, 4.7 mM KCl, 12 mM NaHCO_3_, 0.6 mM KH_2_PO_4_, 1.2 mM MgSO_4_-7H_2_O, 10 mM KHCO_3_, 30 mM Taurine, 5.55 mM Glucose, 500 U heparin, pH 7.4). 6-0 silk was used to tie the aorta around a 22 g metal cannula, which was then affixed to a custom perfusion apparatus and perfused at 37 °C at a constant flow rate of 3–4 mL/min for 5 min. Hearts were then digested by a 6–10 min perfusion of buffer containing 25 µM CaCl_2_ and 2–3 mg Liberase TL until pale and spongy.

Hearts were then removed from the cannula and placed in stopping buffer (18 mL perfusion buffer, 2 mL FBS, 12.5 µM CaCl_2_). The atria were removed and discarded, and the remaining ventricles were gently pipetted into a suspension and filtered through fine nylon mesh. Calcium was slowly reintroduced to the cells in increments of 20 µL of 10 mM CaCl_2_, 20 µL of 10 mM CaCl_2_, 40 µL of 10 mM CaCl_2_, 12 µL of 100 mM CaCl_2,_ and 20 µL of 100 mM CaCl_2_ for a final concentration of 1 mM, with 4 min of incubation at room temperature after each addition. The stopping buffer was aspirated, and the pellet resuspended in 2 mL of Tyrode buffer (135 mM NaCl, 4 mM KCl, 1 mM MgCL_2_, 10 mM HEPES, 1.2 mM NaH_2_PO_4_-H_2_O, 10 mM Glucose, 1.8 mM CaCl_2_, pH 7.4). Cells were plated on laminin-coated, glass-bottom 35 mM dishes and incubated for one hour at room temperature.

### 4.6. FAD/FADH2 Autofluorescence Measurements in Isolated Cardiomyocytes

Plated cardiomyocytes were incubated with either 30 μM Compound A or with the same vehicle volume for 30 min prior to imaging. Live cell imaging was performed with a Zeiss LSM800 confocal microscope (Carl Zeiss, Oberkochen, Germany). Cells were located under the 10× objective with transmitted light, then imaged under the 20× objective with a 405 excitation laser, while emission wavelengths were collected from 520 to 700 nm. Several cells were imaged and analyzed on each plate. Cellular FAD levels were indicated by the level of absolute fluorescence exhibited by cells minus background fluorescence. Cells were monitored without perfusion for 30 s to establish baseline fluorescence. In the first set of experiments, cells were perfused with 1 μM carbonylcyanide-4-trifluoromethoxyphenylhydrazone (FCCP) for 1 min, followed by 1 mM Sodium Cyanide (NaCN) for 5 min. FCCP and NaCN are used to assess the maximally oxidized and reduced pool in order to establish the dynamic range of the FADH_2_ pool and extrapolate a resting “redox index” [53]. In the next set of experiments, cells were perfused with 100 μM diazoxide for 1 min, followed by 1 μM FCCP for 1 min, and finally 1 mM NaCN for 5 min. All perfusion solutions were prepared in the Tyrode buffer described above.

### 4.7. Measurement of Mitochondrial ROS Production in Isolated Cardiomyocytes

Plated cardiomyocytes were treated with 1 μM mitoSOX Red and 30 μM ROMK2 inhibitor Compound A or vehicle. On a Zeiss LSM800 confocal microscope (Carl Zeiss, Oberkochen, Germany), cells were imaged with a 488 nm excitation laser, collecting emission wavelengths from 600 to 700 nm. Thirty seconds of baseline fluorescence was collected before 10–15 min of perfusion or cell death, whichever came first. Cells were perfused with 1 mM H_2_O_2_, 4 μM Antimycin A, a CIII inhibitor, or 10 μM Rotenone, a CI inhibitor. All perfusion solutions were made in the Tyrode Mouse Buffer described above. Cellular ROS levels were indicated by the change in fluorescence over the course of perfusion prior to cell death.

### 4.8. Isolation of Mitochondria

Mitochondria from murine ventricular heart tissue was isolated by homogenization and differential centrifugation, as previously described [37]. Briefly, hearts were quickly excised after thoracotomy and placed in ice-cold isolation buffer (50 mM sucrose, 200 mM mannitol, 5 mM KH_2_PO_4_, 1 mM EGTA, and 5 mM 3-(N-morpholino)propanesulfonic acid). Hearts were manually minced and digested for 10 min with 0.1 mg/mL of trypsin, and mitochondria were isolated with differential centrifugation. The final mitochondrial pellet was resuspended in ice-cold isolation buffer. Mitochondrial protein concentrations were determined by the Bradford method (Bio-Rad, Hercules, CA, USA).

### 4.9. Mitochondrial Membrane Potential Assessment in Isolated Cardiac Mitochondria

Mitochondrial membrane potential (Δψm) was monitored spectrophotometrically as previously described [38,54,55]. Briefly, with the use of rhodamine 123 (5 nM) and excitation λ_ex_ of 503/510 nm and emission λ_em_ of 527/535 nm for coupled and uncoupled mitochondria, respectively, 0.25 mg of mitochondria were added to a cuvette containing malate (5 mM), pyruvate (5 nM) and oligomycin (1 μg/mL). The ionophore FCCP (10 μM) was used to depolarize the mitochondrial membrane at the end of each experiment. The ATP sensitivity of Δψm was assessed after the addition of various concentrations of ATP (1–3000 μM) to a cuvette containing no ATP in the presence or absence of 1 µg of Phosphatidylinositol 4,5-bisphosphate (PIP_2_, Avanti, Birmingham, AL, USA). To investigate the effect of pH on the ATP sensitivity of Δψm, buffers with different pH (range 7–8.2) were used. To assess the effect of ROMK2 inhibition on Δψm and ATP sensitivity, 30 μM of the ROMK inhibitor Compound A [56] was administered in the presence or absence of PIP_2_. Rhodamine 123 fluorescence at each time point was normalized to the fluorescence after the administration of FCCP. Differences between groups were assessed using absolute fluorescence (a.f.u.) changes after drug, vehicle, or ATP administration.

### 4.10. Isolated Cardiac Mitochondria mitoK_ATP_ Swelling Assay

The activity of mitoK_ATP_ was measured spectrophotometrically at 540 nm, as the light scatter (absorbance) changed due to K+ uptake and subsequent swelling in isolated mitochondria [38]. Experimental runs were held within 1.5 h of mitochondrial isolation in a BioTek Synergy H1 Multimode Plate Reader. All wells contained K+ Swelling Buffer (100 mmol/L KCl, 10 mmol/L HEPES, 2 mmol/L MgCl_2_, 2 mmol/L KH_2_PO_4_, pH 7.2 with KOH) in addition to succinate (5 mmol/L) and oligomycin (1 mg/mL). Once the varying test conditions of ATP (0 or 1000 μmol/L) and Compound A (0, 1, 3, 10, 30 μmol/L) were established throughout the wells, isolated mitochondria (0.25 mg/mL) were introduced. Control wells given ATP (1000 μmol/L) included either valinomycin (1 μmol/L) as a positive control or FCCP (10 μmol/L) as a negative control.

### 4.11. Statistical Analysis

Data are reported as mean ± S.E.M. Statistical analyses were performed using the R program 3.4.2 (R Foundation for Statistical Computing, Vienna, Austria) and Microsoft Excel (Microsoft, Redmond, WA, USA) with a *p* value < 0.05 considered significant. Shapiro–Wilk’s test was used to assess normality. Comparisons between 2 groups were made using the two-tailed Student’s *t*-test. Statistical differences between more than two groups were performed by a one-way or two-way ANOVA if assumptions for ANOVA were met. ANOVA was followed by a post hoc Tukey’s test when appropriate.

## 5. Conclusions

In conclusion, our findings demonstrate that pharmacologic ROMK inhibition mediates myocardial IR injury. In addition, our results suggest that ROMK2 may behave preferentially as a mitochondrial K+ efflux channel. With the recent identification of mitoK_ATP_ channel components, the field is ripe for the development of therapeutic agents and clinical translation.

## Figures and Tables

**Figure 1 ijms-26-03795-f001:**
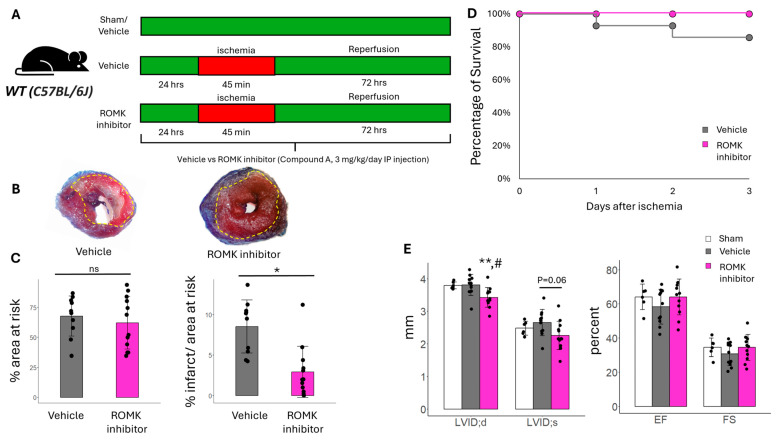
The effect of ROMK inhibition during in vivo IR injury in mice. (**A**) IR injury protocol that subjected mice to 45 min of ischemia and 3 days of reperfusion. The ROMK inhibitor, Compound A, or vehicle was given prior to the ischemic insult and throughout the reperfusion period. (**B**) Representative images of heart slices stained with Evans blue and TTC after IR injury. The yellow dashed line represents the area at risk, and the white solid line represents the infarct area. (**C**) Cumulative data of % area at risk and infarct size after IR injury. (**D**) Survival of mice treated with ROMK inhibition vs. vehicle. Like the ROMK inhibitor treated group, there was no mortality in the Sham arm (not shown). (**E**) Echocardiography after IR injury. LVID; d, left ventricular end diastolic dimension; LVID;s, left ventricular end systolic dimension; EF, ejection fraction; FS, fractional shortening. *N* = 14 mice for vehicle-treated; *N* = 12 mice for ROMK inhibitor-treated. # *p* < 0.05 vs. sham. * *p* < 0.05 vs. vehicle. ** *p* < 0.01 vs. vehicle. Comparisons in (**B**) were made with Student’s *t*-test. One-way ANOVA with the post hoc Tukey HSD test was performed to assess differences between groups in echocardiographic parameters.

**Figure 2 ijms-26-03795-f002:**
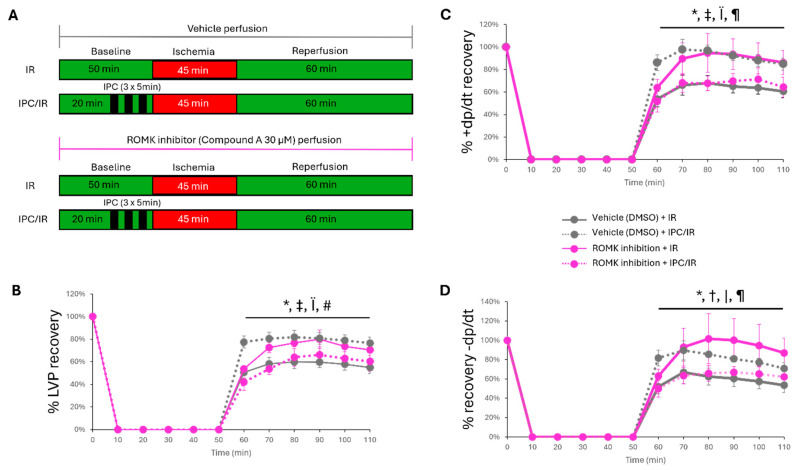
Hemodynamic response during ex vivo IR injury between vehicle and ROMK inhibitor-perfused mouse hearts with and without ischemic preconditioning. (**A**) IR and IPC/IR protocol. Cumulative traces of isolated heart hemodynamic recovery in (**B**) LVDP, (**C**) +dp/dt, and (**D**) −dp/dt. *N* = 7 mice/group; * *p* < 0.001 vehicle IR vs. ROMK inhibitor IR; ^†^ *p* < 0.01, vehicle IR vs. vehicle IPC/IR; ^‡^ *p* < 0.001, vehicle IR vs. vehicle IPC/IR; ^|^ *p* < 0.05 vehicle IPC/IR vs. ROMK inhibitor IPC/IR; ^Ï^ *p* < 0.001 vehicle IPC/IR vs. ROMK inhibitor IPC/IR; ^#^ *p* < 0.01 ROMK inhibitor IR vs. ROMK inhibitor IPC/IR, ^¶^
*p* < 0.001 ROMK inhibitor IR vs. ROMK inhibitor IPC/IR. Two-way ANOVA was performed to assess differences between groups.

**Figure 3 ijms-26-03795-f003:**
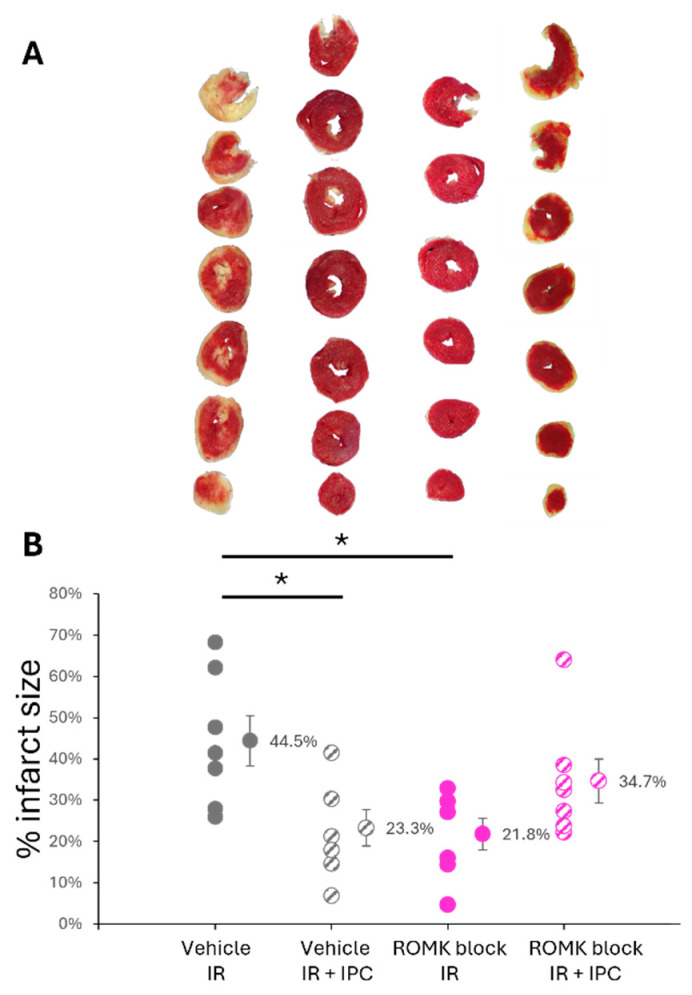
Infarct size after ex vivo IR injury (45 min ischemia, 60 min of reperfusion) between vehicle and ROMK inhibitor-perfused mouse hearts with and without ischemic preconditioning. (**A**) Representative heart images. (**B**) Cumulative % infarct data. *N* = 7 mice/group; * *p* < 0.05. One-way ANOVA was performed with the post hoc Tukey HSD test to assess differences between groups.

**Figure 4 ijms-26-03795-f004:**
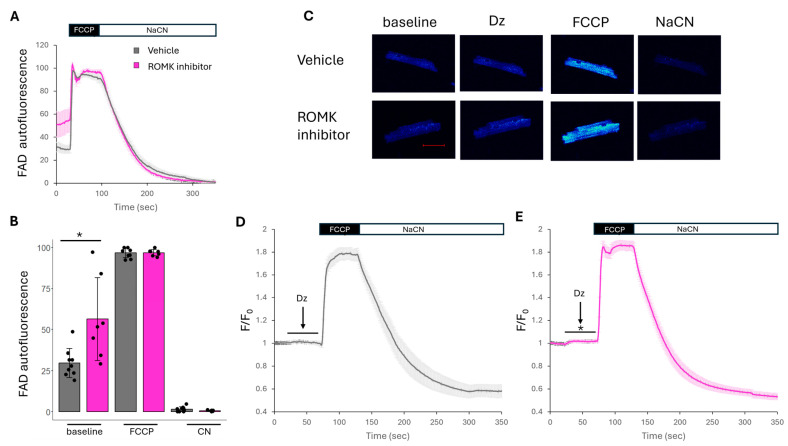
The effect of ROMK inhibition on FAD autofluorescence. (**A**) Cumulative traces of FAD autofluorescence from isolated murine cardiomyocytes pretreated with and without a ROMK inhibitor subsequently perfused with 1 μM FCCP followed by 1 mM NaCN. (**B**) Cumulative data of baseline FAD autofluorescence between vehicle and ROMK-treated cells. (**C**) Representative images of isolated cardiomyocytes treated with and without a ROMK inhibitor perfused with 100 µM Diazoxide followed by 1 μM FCCP and 1 mM NaCN; scale bar = 50 μm. (**D**,**E**) Cumulative traces of FAD autofluorescence in vehicle or ROMK inhibitor-treated cardiomyocytes perfused with 100 µM Diazoxide followed by 1 μM FCCP and 1 mM NaCN. For (**A**,**B**), *N* = 8 for vehicle-treated and *N* = 7 for ROMK inhibitor-treated. For (**D**,**E**), *N* = 19 cells for vehicle-treated, *N* = 21 cells for ROMK inhibitor; * *p* < 0.05. Comparisons in (**B**) were made with Student’s *t*-test. Two-way ANOVA was performed to assess differences during diazoxide perfusion.

**Figure 5 ijms-26-03795-f005:**
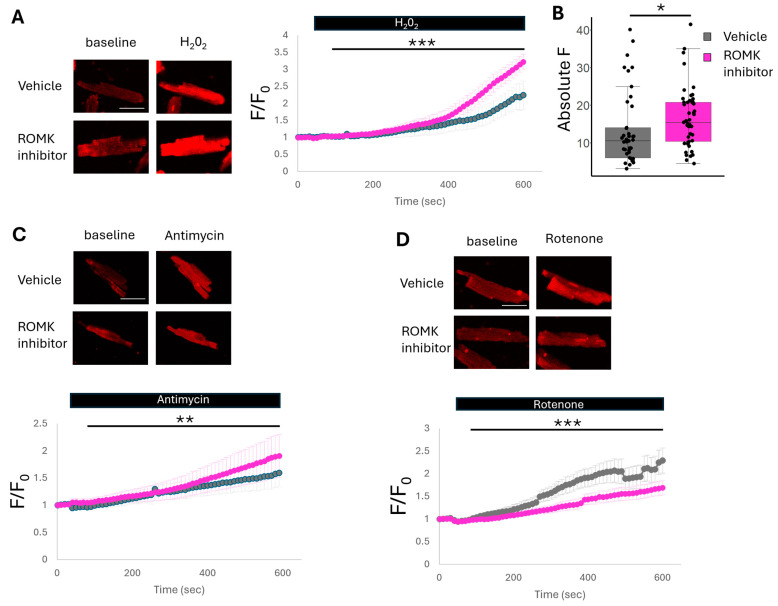
Mitochondrial superoxide generation in isolated cardiomyocytes with and without ROMK inhibition. (**A**) Representative images and cumulative data of isolated adult murine cardiomyocytes stained with mitoSOX red and treated with vehicle or ROMK inhibitor (30 μM Compound A) before and after perfusion with 1 mM H_2_O_2_. (**B**) Absolute mitoSOX fluorescence of all cells at baseline prior to perfusion treated with vehicle or ROMK inhibition. (**C**) Representative images and cumulative data of isolated adult murine cardiomyocytes stained with mitoSOX red and treated with vehicle or ROMK inhibitor (30 μM Compound A) before and after perfusion with 4 μM Antimycin. (**D**) Representative images and cumulative data of isolated adult murine cardiomyocytes stained with mitoSOX red and treated with vehicle or ROMK inhibitor (30 μM Compound A) before and after perfusion with 10 μM Rotenone. For panels (**A**,**C**,**D**) pink cumulative line traces are ROMK inhibitor treated cells while gray lines represent vehicle treated cells. *N* = 4 mice; for H_2_O_2_, *N* = 11 cells for vehicle, *N* = 13 cells for ROMK inhibition; for Rotenone, *N* = 12 cells for vehicle, *N* = 19 cells for ROMK inhibition; for antimycin, *N* = 12 cells for vehicle, *N* = 11 cells for ROMK inhibition; scale bar = 50 μm; *** *p* < 0.001, ** *p* < 0.01, * *p* < 0.05. Comparisons between groups in (**B**) were made with Student’s *t*-test. Two-way ANOVA was performed to assess differences in perfusion experiments.

**Figure 6 ijms-26-03795-f006:**
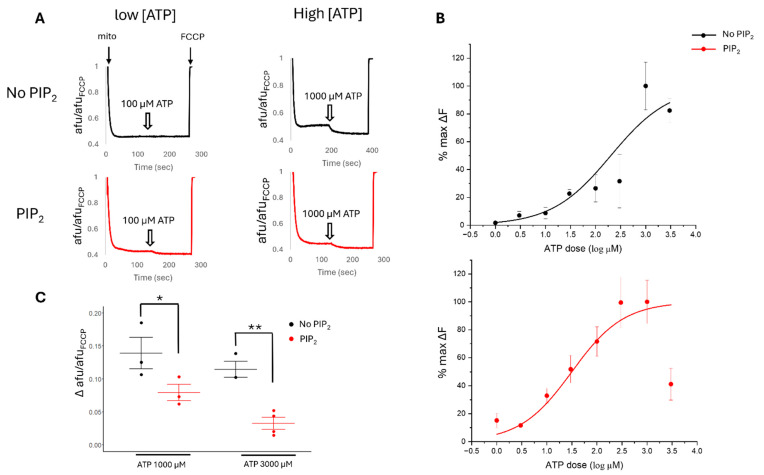
The regulation of mitochondrial membrane potential (ψm) by ATP in isolated mouse heart mitochondria. (**A**) Representative traces of the effect of low (100 µM) or high (1000 µM) ATP on ψm in the presence or absence of 1 µg PIP_2._ (**B**) Dose–response curves of Δψm to different [ATP] with and without 1 µg of PIP_2_. (**C**) The effect of high doses of ATP on ψm in the presence or absence of 1 µg PIP_2_. *N* = 3–5 trials/group. * *p* < 0.05, ** *p* < 0.01.

**Figure 7 ijms-26-03795-f007:**
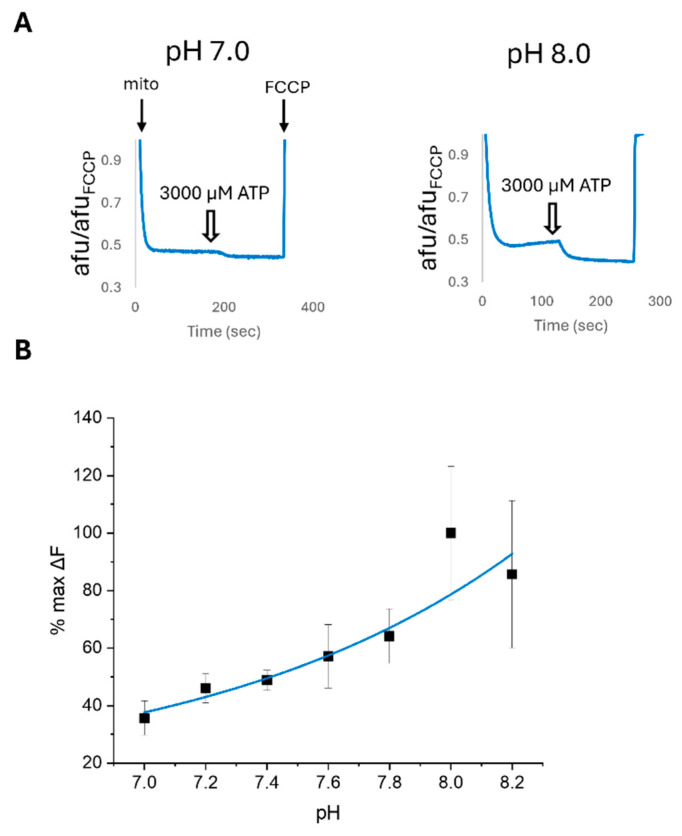
The effect of pH on the regulation of mitochondrial membrane potential (ψm) by ATP in isolated mouse heart mitochondria. (**A**) Representative traces of the effect of 3000 µM ATP on ψm at a pH of 7.0 vs. 8.0. (**B**) Dose–response curve of Δψm treated with 3000 µM ATP at different pH levels. *N* = 3–5.

**Figure 8 ijms-26-03795-f008:**
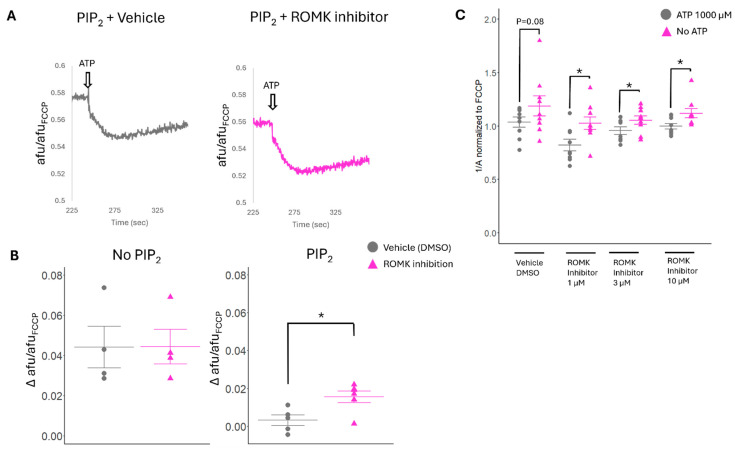
Exaggerated effect of ATP hyperpolarization on ψm after ROMK inhibition. (**A**) Representative traces of the effect of 1000 µM ATP on ψm with or without ROMK inhibition in the setting of PIP_2_. (**B**) Cumulative data of Δψm after 1000 µM ATP treatment with and without ROMK inhibition and PIP_2_. (**C**) Mitochondrial swelling measured by light scattering at 540 nm using a spectrophotomer normalized to swelling induced by mitochondria treated with FCCP. Conditions either had the presence of 1000 µM of ATP or the absence of ATP. The effect on mitochondrial swelling of different concentrations of the ROMK inhibitor, Compound A (0, 1, 3, 10 µM), was assessed. *N* = 4–6 trials/group for ψm experiments; *N* = 9 trials/group for swelling experiments; * *p* < 0.05. Comparisons between groups were made with Student’s *t*-test.

**Figure 9 ijms-26-03795-f009:**
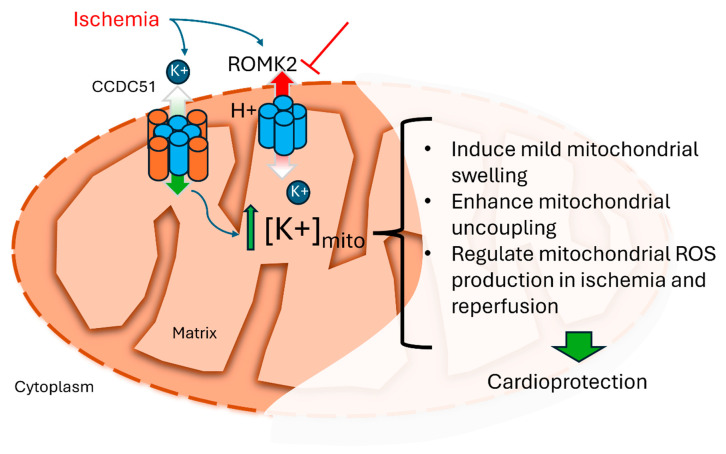
Proposed hypothesis for cardioprotection from inhibition of cardiac mitochondrial ROMK. Increased K+ in the mitochondrial matrix leads to several observed events that are associated with the prevention of ischemic injury. CCDC51 serves primarily as a mitochondrial K+ influx channel. The inherent properties of ROMK2 as an inward rectifying K+ channel and conditions during ischemia may promote ROMK2 to behave as a K+ efflux channel, which includes pH changes, increased matrix [K+], mitochondrial membrane potential depolarization, and activation of protein kinases. The blockade of ROMK2 during ischemia, we propose, increases mitochondrial matrix K+. We hypothesize that the inhibition of ROMK2 represents a novel therapy in ischemic heart disease.

**Table 1 ijms-26-03795-t001:** Baseline cardiac function in ex vivo perfused hearts with and without ROMK inhibition.

	WT Vehicle (*N* = 7)	WT Compound A (*N* = 7)	*p* Value
LVDP, mmHg	61 ± 5.9	54 ± 7.6	0.23
+dp/dt, mmHg/s	1947 ± 198	1590 ± 214	0.12
−dp/dt, mmHg/s	−1339 ± 174	−1024 ± 183	0.11
HR, bpm	310 ± 16	247 ± 44	0.1

## Data Availability

Data are available on request to the corresponding author.

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
