# Peer review of "Pharmacologic ROMK Inhibition Protects Against Myocardial Ischemia Reperfusion Injury"

_ijms, 2025, doi:10.3390/ijms26083795_

Round 1

Reviewer 1 Report

Comments and Suggestions for Authors

The study is very interesting and raises the important issue of ischemia-reperfusion injury and cardioprotection. The results are an important voice on the topic and the basic science. The study seems to have been conducted correctly and I do not see any methodological or statistical errors. I have some minor comments:

Methods to reduce the extent of the infarction are still being sought. One method is the use of substances that are administered to patients during ischemia (transport to the hospital), immediately before or during reperfusion (percutaneous coronary intervention). Administering substances for 24 hours before myocardial injury is not an option. Therefore my only doubt about the study arises from the possibility of translating it to the clinic – what caused the ROMK inhibitor to be administered for so many hours? And my comment concerns the first sentence in the conclusion: ‘In conclusion, our findings demonstrate that ROMK inhibitors represent a novel pharmacologic approach to treat myocardial IR injury” - in my opinion, the conclusion from the study  is different (should focus on the basic science) and is too far advanced. This sentence may be used, however, not as the first and main conclusion.

The abbreviation HTN was used once without explanation line 71

Reviewer 2 Report

Comments and Suggestions for Authors

The manuscript “Pharmacologic ROMK inhibition protects against myocardial ischemia reperfusion injury” by Wexler et al. addresses a very interesting role of a K+-channel in mitochondria. The study highlights the ROMK channel as a potential therapeutic target for cardiac pathologies associated with ischemia/reperfusion.

The manuscript is well written, experiments conducted with appropriate controls and number of replicates. I have only minor points to address:

  1. In the introduction I would suggest adding a sentence about the levels of K+ ions in mitochondrial matrix, mainly in comparison to the cytosolic levels.
  2. In Figure 1 it is not clear how did authors define the infarcted area and the area at risk to calculate the percentage. Thus, I recommend delineating these two areas on the two images included in panel B.
  3. OY axis titles are very small in general, and in Fig 1E in particular. Please increase the font size.
  4. In Figure 3, for both the legend and OY title – correct the infract and infratc respectively
  5. Please bring references regarding the level of PIP2 in inner mitochondrial membrane, when first mentioned, L197, or under Discussions
  6. Explain the parameter “afu” used on the OY axes of Figure 6 A, C or 7A
  7. In fig 6A, the panel corresponding to no PIP2 and high ATP is slightly smaller.
  8. Figure 8A – use the same interval for OY, such as 0.5 to 0.6; same for 8B.
  9. L290 correct form to from, maybe?
  10. L326: mitochodria has a high positive potential.
  11. L433 – please correct the unit for the transverse slices.
  12. OY title of Fig S2 - specify that it is the mitochondrial membrane potentia
